# Ecological and Environmental Benefits of Planting Green Manure in Paddy Fields

**Beining Lei [1,2], Juan Wang [2,3] and Huaiying Yao [1,2,3,*]**

1   Research Center for Environmental Ecology and Engineering, School of Environmental Ecology and Biological Engineering, Wuhan Institute of Technology, Wuhan 430073, China; 21916010015@stu.wit.edu.cn
2   Zhejiang Key Laboratory of Urban Environmental Processes and Pollution Control, Ningbo Urban Environment Observation and Research Station, Chinese Academy of Sciences, Ningbo 315800, China; jwang@iue.ac.cn
3   Key Laboratory of Urban Environment and Health, Institute of Urban Environment, Chinese Academy of Sciences, Xiamen 361021, China
*   Correspondence: hyyao@iue.ac.cn

**Abstract:** Soil fertility management is one of the most important factors affecting crop production. The use of organic manures, including green manure, is an important strategy to maintain and/or improve soil fertility for sustainable crop production. Green manure generally refers to crops that can provide fertilizer sources for agricultural cash crops and improve soil productivity. The application of green manure is a traditional and valuable practice for agroecosystem management, particularly in paddy systems where green manure is rotated with rice. This paper systematically reviews the effects of green manure on soil microenvironments and greenhouse gas emissions, and the role of green manure in the phytoremediation of paddy fields. The paper concludes that green manure can not only affect soil nutrients and the microbial community, but also reduce greenhouse gas emissions and enhance soil remediation to some extent. Moreover, this review provides theoretical guidance on the selection of green manure germplasm and tillage methods for paddy fields of different climates and textures. However, this review only provides a macro-overview of the effects of green manure on soil nutrients, greenhouse gas emissions, and soil remediation in rice paddies based on a large number of previous studies, and does not provide a comprehensive quantitative assessment due to differences in green manure varieties and soil texture. The prospects for quantitative analysis of the ecological and economic effects of the sustainable development of green manure cultivation are discussed.

**Keywords:** green manure; soil microenvironment; greenhouse gas; phytoremediation; environmental benefit

## 1. Introduction

Green manure crops generally refer to crops that can serve as nutrients for economic agricultural crops and improve soil productivity; this mainly includes legumes that can fix nitrogen by rhizobia, such as Chinese milk vetch and alfalfa, and non-legumes, such as ryegrass and barley. Leguminous green manure crops were recognized as an important source of N for wetland rice well before the advent of modern agricultural technology in China, India, and Japan. Some common varieties of green manure are collated in Table 1. According to incomplete statistics, the planting area of green manure gradually decreased from 1950 to 1980 in China [1], which may have been caused by the development of a social economy and the transformation of agricultural production modes. Chemical fertilizer has gradually become the main source of nutrients for crop growth [2]. However, the advantages of green manure have rapidly become apparent due to the soil fertility decline, farmland nonpoint source pollution and soil acidification caused by the long-term application of chemical fertilizers, the introduction of ecological agriculture service projects [3], and the deepening understanding of green manure by the public in recent years. More importantly, under the guidance of "carbon peak" and "carbon neutral"

policies in agricultural land, the practical application of green manure has great significance to ensure national food security and protect the ecological environment associated with agricultural production.

**Table 1.** The most common green manure crops [4–8].

| Condition | Classification | Species Examples |
|---|---|---|
| Climate | Temperate continental climate; Monsoon climate of medium latitudes; Plateau mountain climate | *Orychophragmus violaceus, Vicia villosa, Brassica campestris, Melilotus officinalis, Lolium perenne, Vicia sepium, Secale cereale, Avena sativa* |
| | Subtropical monsoon climate; Subtropical climate | *Carthamus tinctorius, Chinese milk vetch, Trifolium repens, Ryegrass, Hairy vetch, Vicia faba, Cassia tora, Chinese pennisetum, Crotalaria mucronata* |
| Growth cycle | Annual | *Field pea (Pisum sativum* L.), *Black lentil (Lens culinaris Medikus), Chickling vetch (Lathyrus sativus* L.), *Fababean (Vicia faba* L.), *Hairy vetch (Vicia villosa Roth* ssp. *villosa), Woolypod vetch (Vicia villosa Roth* ssp. *varia), Tangier flatpea (Lathyrus tingitanus* L.), *Annual medics (Medicago* spp.), *Berseem clover (Trifolium alexandrinum* L.) |
| | Biennial or Perennation | *Sweet clover (Melilotus officinalis* L.), *Red clover (Trifolium pratense* L.), *Melilotus officinalis, Trifolium repens* |
| Family | Cereal crop | *Lolium multiflorum, Fagopyrum esculentum, Secale cereale, Triticum aestivum, Hordeum vulgare, Avena sativa, Ryegrass* |
| | Brassica crop | *Brassica napus, Brassica rapa, Raphanus sativus* |
| | Leguminosae crop | *Trifolium alexandrinum, Vigna unguiculata, Trifolium incarnatum, Vicia villosa, Trifolium pretense, Melilotus officinalis, Trifolium repens* |

Rice serves as a staple food for more than 60% of the population in China, and reasonable crop rotation of green manure and rice has important advantages in reducing the application of chemical fertilizers and improving soil fertility and rice quality. In recent years, China has introduced subsidy policies for green manure cultivation, proposed the concept of sustainable ecological development, and launched relevant soil organic matter enhancement projects, accelerating the widespread application of green manures in rice production. Figure 1 illustrates the role of green manure in the rotation with rice for nutrient supplies, greenhouse gas emissions, and soil remediation in paddy fields. Although this review discusses trends in the efficient utilization of green manure resources in paddy ecosystems, it does not provide a comprehensive quantitative analysis and only provides a theoretical basis and guidance for the sustainable development of green manure return in rice production.

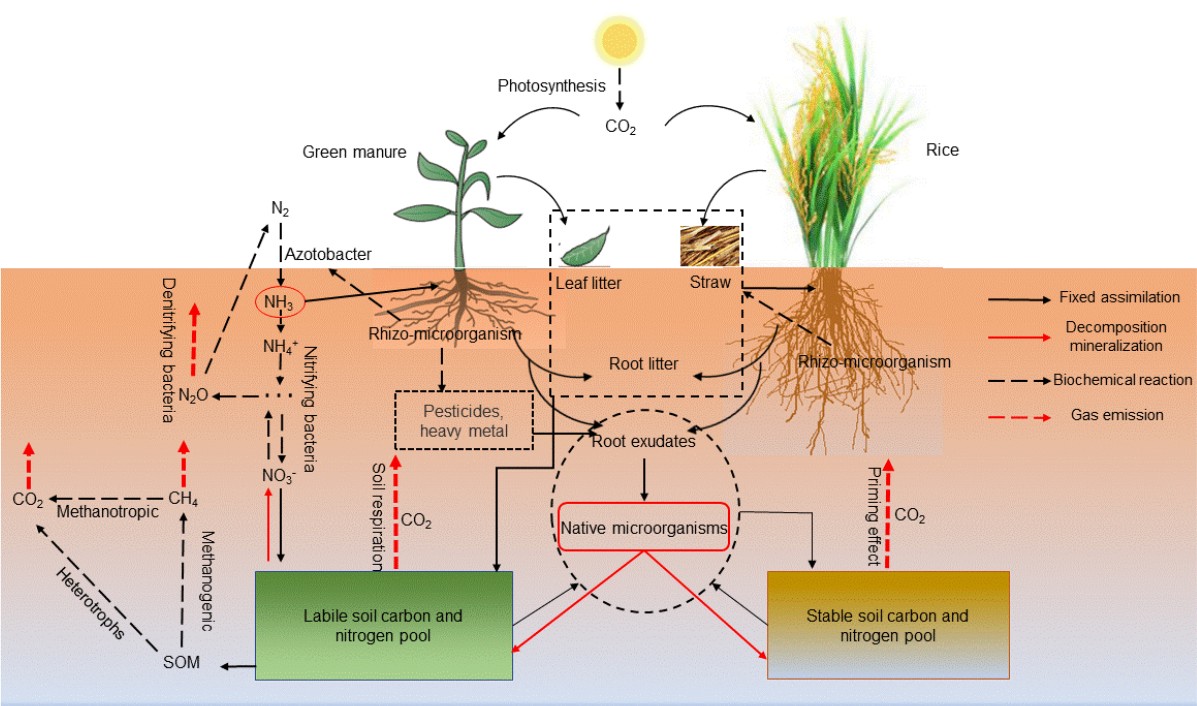

**Figure 1.** Schematic diagram of carbon and nitrogen cycles and functions of green manure in a green manure and rice rotation system.

## 2. The Effects of Green Manure on the Paddy Soil Microenvironment

### 2.1. Physical and Chemical Properties

Green manure can promote soil health, reduce soil erosion [9], improve soil hydraulic properties, and increase soil organic carbon pools [10]. Natural soil erosion, which consists mainly of wind and water erosion, is a major factor in soil erosion. Wind erosion of agricultural soils occurs mainly in arid and semiarid areas and is mainly due to dust and sand [11]. Green manure planting reduces soil exposure, improves soil water-holding capacity, and increases the content of soil aggregates, thus reducing soil's wind erosion [12]. The water erosion of agricultural soils mainly occurs in hilly areas with high rainfall and is mainly caused by the gradual loss of surface fertile soil to water run-off. Planting green manure can reduce surface runoff and flow rates and prevent surface soil erosion. However, paddy fields are generally located in plain areas and, during the winter season, high latitude paddy soils are vulnerable to wind erosion. Planting green manure can reduce erosion by lowering daytime soil temperatures, reducing evaporation, and increasing soil permeability to maintain soil water content [13]. In soil tectonics, infiltration is one of the main internal erosion processes, selectively eroding the fine particles that move through the voids formed by the coarser particles [14]. Green manure planting in high-latitude paddy fields can reduce soil consolidation caused by rainy seasons by increasing the content of water-stable aggregates and soil permeability [15], which is related to the soil colloid content caused by green manure root secretions and microbial enzyme activity [16]. Low-latitude rice fields are susceptible to water erosion caused by heavy rainfall during the winter season, and green manure varieties with a well-developed root system should be selected to protect the soil from nutrient loss in the topsoil. In addition, green manure acts as a good filter for both root microorganisms and leachable nutrients, and for the fertilizer and symbiotic organisms added to the topsoil [17]. The effects of green manure on crop growth and nutrient utilization are related to the improvement in soil physical and chemical properties, including soil bulk density, water conductivity, and carbon and nitrogen levels [18]. Previous studies discovered that long-term application of green manure could reduce soil bulk density and the soil organic carbon consumption rate, and improve

soil porosity, water-holding capacity, enzyme activity, organic matter, total nitrogen, and available nitrogen contents [19,20]. At the same time, green manure can improve the content and stability of soil water-stable aggregates and enhance the resistance of soil microorganisms to environmental stress. Ma et al. [21] showed via a meta-analysis that green manure can effectively improve soil quality, reduce soil bulk density by approximately 5.6%, increase microbial biomass carbon by approximately 28%, and improve soil enzyme activity by approximately 14–39%.

Green manure varieties are abundant, and the fertility characteristics and production capacity of different varieties of green manure vary greatly. Therefore, different tillage patterns should be adopted according to the fertility of green manure and the characteristics of the variety, combined with climatic and soil texture conditions. The world's major rice producing regions are concentrated in East Asia and Southeast Asia, where the climate types are mainly tropical monsoon, subtropical monsoon, and temperate monsoon. Several common green manure species are organized according to their climatic classification, as shown in Table 1. The main types of paddy soil textures with low fertility are acidic red soil and alkaline black soil [22]. Influenced by tropical and subtropical climatic conditions, the high temperatures and heavy rainfall in red soil areas synchronize with strong weathering and leaching to make the red soil naturally low in fertility. A previous study found that long-term planting of milk vetch and rape as green manures increased the degree of aromaticity, humification, and average molecular weight of dissolved organic matter (DOM), and made the DOM more stable in red paddy soil [23]. Moreover, planting milk vetch and ryegrass can improve the availability of structural potassium (K) and K retention capacity in the soil in relation to soil mineralogy, and thus can be an effective alternative to enhance soil K availability in red paddy soil [24,25]. Hong et al. [26] suggested that long-term rice–rice–green manure rotation can significantly change the apparent nitrogen and phosphorus balance and their association with soil nitrogen and phosphorus content, respectively. Alkaline black soils affect rice production due to their high salt content. Shirale et al. [27] found that planting *dhaincha* and *sunhemp* in sodic soils enhanced nutrient supply and showed potential to reclaim sodic soils [28]. In addition, fertilizer application methods and tillage practices also have a significant effect on soil nutrients. Li et al. showed that the use of green manure and straw for fertilizer systems of grain crops helps to increase the return of nutrients to the soil. At the same time, the content of mineral nitrogen increased by 5.9 mg kg$^{-1}$, mobile phosphorus by 21 mg kg$^{-1}$, and exchangeable potassium by 14.5 mg kg$^{-1}$ of soil, on average, during the growing season in the arable layer of typical black soil [29]. Integrating green manure with conservation tillage (reduced-tillage and no-tillage) systems can improve agroecosystem performance even within a short time. No-till cover cropping accumulated greater microbial biomass C and available phosphorous (P) and K in the 0–10 cm soil depth than intensive tillage in a semiarid agroecosystem in Denmark [30]. In another study, lentil green manure under no-tillage produced 9.1% greater SOC in the top 15 cm of soil compared with soil incorporation of the lentil crop by disking [31].

Soil pH is an important factor of soil health in paddy fields, and soil acidification is one of the main problems facing agricultural soils in China [32]. The application of green manure during the rice growing process will mostly acidify the soil [33]. The organic acids released during straw decomposition can significantly reduce soil pH and soil redox potential, and then lead to soil acidification [34,35]. However, this phenomenon is not conducive to the growth of rice roots and inhibits the tillering of rice. Moreover, Gao et al. [36] found that long-term rice–rice–green manure rotation increased the microbial richness and diversity, and shifted the community composition and structure, particularly decreasing soil acidification with more sulfate-reducing bacteria and fewer sulfur-oxidizing bacteria. In conclusion, the reduction in soil pH caused by applying green manure is a short-term effect caused by the rotting of the green manure crop, whereas the change in the soil microbial community caused by long-term rotation of green manure causes a long-term

soil pH increase. Therefore, attention should be given to the positive effects of long-term green manure planting on alleviating soil acidification.

*2.2. Nitrogen Utilization*

Nitrogen is the most important limiting nutrient in paddy field ecosystems. Both chemical fertilizer and green manure can increase the efficiency of nitrogen availability in the process of rice growth by promoting the nitrogen cycle in the rice–soil–microbial system [37]. Islam et al. [38] showed that replacing chemical fertilizer with green manure can improve the growth environment of rice roots and increase root density, biomass, activity, and nutrient absorption. In addition, legumes such as green manure can provide additional nitrogen for the soil [39]. Gao et al. found that the relative abundance of rhizobia in soil with long-term green manure in rotation with rice was higher than in the winter fallow treatment based on a long-term field experiment that involved four rotation systems, namely, rice–rice–winter fallow, rice–rice–ryegrass, rice–rice–rape, and rice–rice–milk vetch, which indicated that the long-term application of green manure can significantly increase the soil total nitrogen content in paddy fields [36]. The isotope labelling technique has also been widely used to study the decomposition and nutrient release of green manure. Zhu et al. [40] studied the nitrogen release process of $^{15}$N-labelled green manure (Chinese milk vetch) during the growth of double-season rice and found that approximately 39–46% of green manure nitrogen was absorbed by rice, and approximately 29–33% remained in the soil after 177 days of green manure application.

The improvement in crop nitrogen utilization efficiency depends largely on the synchronization of crop nitrogen demand and nitrogen supply from different sources throughout the growing season [41]. The combined application of green manure and chemical fertilizer can improve the soil nitrogen supply capacity by promoting the mineralization potential and rate of soil organic nitrogen, and then promoting the absorption and accumulation of nitrogen by rice [42]. The combined application of milk vetch and urea can promote rice growth and increase the utilization rate of urea nitrogen [43,44]. Green manure also affects gene abundance related to nitrogen cycling of microorganisms in rice fields [45]. Gao et al. [46] discovered that the application of Chinese milk vetch significantly increased the relative abundance of soil ammonia-oxidizing archaea (AOA) and ammonia-oxidizing bacteria (AOB), which may be the result of increased nitrogen availability in soil caused by mineralized nitrogen in green manure plant residues [47]. The application of green manure provided a more stable form of nitrogen supply and reduced the nitrification potential by reducing nitrification substrates, which maintained a relatively stable trend in the $NH_4^+$-N content in the soil [48]; consequently, it reduced soil nitrogen volatilization, leaching and runoff, which was conducive to the protection of the aquatic and atmospheric environment.

The dry matter accumulation and nitrogen uptake of rice vary with different growth stages after green manure application [49]. The dry matter accumulation of early rice was found to decrease from the seedling stage to the tillering stage but increased linearly from the heading stage to the maturity stage, so the dry matter accumulation of all treatments after green manure application was higher than that of the control. The dry matter accumulation of late rice was significantly higher than that of the control during the whole growth period, suggesting that green manure application may have a long-term and sustainable effect on soil fertility [50]. In addition, green manure return results in a rapid increase in the number of soil microorganisms, but the large C/N and C/P ratios of non-leguminous green manure crops increase the uptake of available N and P in soil by microorganisms, which reduces the contents of available N and P in soil at the initial stage of green manure return and then inhibits the early growth of rice [51]. Green manure plants contain major macronutrients (N, P, K) and more microelements (Ca, Mg, Si), which can not only transfer nutrients to the soil and ensure the balanced and sustainable supply of nutrients in paddy soil, but also promote the absorption of nutrients from deeper soil by rice roots [52–54]. In conclusion, applying green manure is an effective method to improve nitrogen management and increase rice yield.

*2.3. Microbiome*

The application of green manure also has a considerable effect on the soil microbial community at the regional scale. Compared with other organic fertilizers, one of the remarkable characteristics of green manure is that its growth process can affect paddy soil and can cause different responses of the corresponding microorganisms. Green manure–rice rotation can affect the microbial community structure and nutrient cycling, especially for organisms related to soil carbon (C), nitrogen (N), and sulfur (S), which are methane-oxidizing bacteria, sulfur-reducing bacteria, and functional bacteria related to nitrogen fixation, nitrification, and denitrification [36,55]. In particular, the application of Chinese milk vetch increased the relative abundance of actinomycetes and Firmicutes in paddy soil, which was associated with actinomycetes promoting the decomposition of plant residues [56]. Zhang et al. [57] isolated 169 strains and 77 strains of endophytic bacteria from rice roots under green manure–rice and rice–winter fallow cultivation modes, respectively. 16S rRNA gene analysis showed that the 169 strains of the former were divided into 19 genera and 21 species, whereas the 77 strains of the latter were divided into 14 genera and 15 species. These results indicated that long-term green manure–rice rotation significantly increased the diversity of the endophytic bacterial community in rice roots, and the endophytic bacteria of the rotation were more beneficial to rice growth than those of the rice monoculture. In conclusion, green manure–rice rotation significantly affected the soil microbial community structure, increased the soil microbial biomass, and changed the diversity of endophytic bacteria in rice roots, which was more conducive to the absorption and utilization of nutrients by rice.

## 3. The Effects of Green Manure on Greenhouse Gas Emissions from Paddy Fields

The increase in the concentration of greenhouse gases in the atmosphere is one of the main causes of global warming, among which the contribution rates of $CO_2$, $N_2O$, and $CH_4$ to the greenhouse effect are 76.7%, 7.9%, and 14.3%, respectively [58]. Paddy soil is the main source of greenhouse gas emissions, accounting for approximately 10–12% of the total global agricultural emissions sources [59]. To compare rice yield and greenhouse gas emissions in field experiments, many researchers use the total greenhouse gas flux released per unit of rice yield as an indicator of the effects of rice cultivation on global warming [60]. In rice rotation systems, agricultural management measures such as organic matter inputs, mineral fertilizer application, and irrigation will affect the greenhouse gas emissions of rice soil [61], but the impact of green manure on greenhouse gas emissions has rarely been systematically investigated.

*3.1. $CO_2$ Emissions*

It was found that most of the green manure carbon was converted to $CO_2$ and released in the process of decomposition, whereas only a small amount of green manure carbon was converted to soil organic carbon [62,63]. In the soil of the green manure–rice rotation, the release of $CO_2$ is mainly due to soil respiration and the result of the priming effect produced by green manure residue. According to the mechanism of soil $CO_2$ production, soil respiration is usually divided into microbial heterotrophic respiration and root respiration [64]. In green manure–rice rotation soil, to evaluate the environmental benefits of green manure, heterotrophic respiration is generally considered. The process of green manure planting and return provides root exudates and fresh carbon inputs to the rice soil, which improves the availability of the substrate for microbial respiration [65], thereby increasing the intensity of soil respiration. Water content and oxygen availability in paddy soil are also key factors affecting the soil respiration response. During the growth stage of green manure, the paddy soil is in the non-flooded period. At this time, the soil oxygen availability is high, and the soil water content is appropriate, which is conducive to soil respiration and carbon release. The soil respiration rate will also increase with increasing soil temperature [66]. This response may exceed the fixed carbon produced by plants and cause net carbon loss in the soil [67]. Zheng et al. [68] discovered that green manure and chemical

fertilizers combined with powder ridge tillage reduced the intensity of soil respiration during the no-tillage and full heading and harvest stages of late rice and was beneficial to reducing the release of soil $CO_2$. Carter M.S. et al. [69] showed that in the first week after the application of green manure, the $CO_2$ emissions caused by green manure were equivalent to 72–79% of the dissolved organic carbon of green manure, and within three months of green manure application, approximately 32–54% of green manure carbon was released by respiration, indicating that the active component was completely assimilated or decomposed in the microbial biomass within a few days. Therefore, despite the $CO_2$ release caused by heterotrophic respiration, 20–30% of the carbon is fixed in the paddy soil in the form of green manure, which is of great significance to the maintenance of the organic carbon pool of the paddy soil.

The positive priming effect caused by green manure crops as an external carbon input to mineralize the organic carbon of paddy soil will also cause the release of $CO_2$ in paddy soil [70]. An increase in green manure return can improve the amount of accumulated carbon mineralization in the soil of powder ridge paddy fields, stimulate the potential mineralizable organic carbon and mineralization rate of the soil, and increase the soil respiration intensity of early rice [68]. However, Xu et al. [71] found that the combined application of green manure and chemical fertilizers reduced $CO_2$ emissions from paddy fields compared with those under chemical fertilizers alone, and the cumulative $CO_2$ emissions of conventional no-tillage paddy fields with green manure combined with chemical fertilizers were significantly lower than those of powder-ridge no-tillage paddy fields. This may be because green manure has an impact on the soil microenvironment of the no-tillage paddy soil after the stable transformation in the paddy field, and the release of $CO_2$ is slowed. In addition, plowing green manure in the field releases more $CO_2$ than mulching in the field [72]. This may be due to the greater disturbance of the soil, improved soil aeration, and enhanced respiration intensity of microorganisms by plowing [73].

*3.2. CH$_4$ Emissions*

$CH_4$ is mainly produced by methanogenic archaea decomposing organic matter under anaerobic conditions. The microbial decomposition rate of organic matter in paddy soil and the transport capacity of methane through aerated tissue are important factors that determine the rate of $CH_4$ emissions [74]. The application of green manure can consume oxygen in the soil in advance, thereby reducing the redox potential of the soil and ultimately leading to an increase in $CH_4$ emissions [75]. Compared with the sole application of chemical fertilizers, the combined application of green manure with chemical fertilizers promoted $CH_4$ emissions from paddy fields, which may have been due to the decomposition of green manure that provided rich substrate nutrients for the growth of methanogens in paddy fields. The combined application of ryegrass with a high C/N ratio and chemical fertilizer caused a significantly higher $CH_4$ flux than the combined application of Chinese milk vetch with a low C/N ratio and chemical fertilizer [76]. However, Zhong et al. [77] found through two-year field experiments that Chinese milk vetch and rice rotation significantly reduced the annual cumulative emissions of $CH_4$ (33–63%). The main reason was that milk vetch reduced the soil C/N ratio, thereby reducing the abundance of methanogens and increasing AOA abundance and soil permeability.

In addition to reducing $CH_4$ emissions by changing the types of green manure and fertilization methods, Song et al. [78] discovered that aerobic short pre-digestion (10–20 d) before green manure return can significantly reduce $CH_4$ flux during rice planting (55–80%), and the underlying mechanism is to convert exogenous unstable carbon into $CO_2$ with a lower warming potential through aerobic digestion, thereby reducing $CH_4$ emissions. However, during the fallow stage of the paddy field, with the extension of the short aerobic pre-digestion period, $N_2O$ emissions from the paddy field can be significantly increased. By comparison, after 30 days of short aerobic pre-digestion, the impact of the environmental net carbon balance on the warming potential is greater than the impact on

$CH_4$ emissions from paddy fields, so this factor should be controlled within 30 days to optimize environmental benefits.

*3.3. $N_2O$ Emissions*

Nitrogen cycling processes in soil mainly include nitrification, denitrification, organic N mineralization, and $N_2$ fixation [79]. $N_2O$ is produced in the process of soil microbial nitrification and denitrification in paddy fields [80]. The microbial conversion process and nitrogen utilization efficiency of nitrogen fertilizer in the soil vary with the type of fertilizer. Nitrogen release after applying green manure is affected by soil moisture, temperature, the C/N ratio, and the concentration of highly resistant components such as lignin [81]. The combined application of green manure and chemical fertilizers significantly increased the $N_2O$ emissions of early rice and late rice [82]. However, at the same level of nitrogen application, the $N_2O$ emissions of a single application of chemical fertilizer were higher than those of a single application of green manure and green manure combined with chemical fertilizers. The reason may be related to the higher C/N ratio of ryegrass, which in turn leads to soil nitrogen fixation [83]. In addition, shallow burying of green manure crops in the soil is more conducive to reducing the release of $N_2O$ than deep burying in the soil.

During the flooding period of rice, extremely reducing conditions inhibited nitrification, resulting in denitrification being more marked for $N_2$ than for $N_2O$ or $NO$ [78]. Maintaining the soil moisture level at 36–39% can minimize the loss of N caused by denitrification [69]. However, the release of $N_2O$ induced by green manure mainly occurs before the net release of N derived from green manure, which indicates that $N_2O$ is mainly derived from the denitrification of the initial $NO_3^-$-N in the soil, and in the improvement of soil by green manure, nitrification is the secondary source of $N_2O$. Another possible mechanism for the increase in $N_2O$ release in paddy fields is that the lowering of local soil pH when green manure decomposes increases the $N_2O/N_2$ product ratio of denitrification. Local acidification of the soil may increase the ratio of fungi/bacteria, which in turn increases $N_2O$ release [84]. Zhong et al. [77] verified this hypothesis and showed that Chinese milk vetch and rice rotation increased the cumulative emission of $N_2O$ (17–870%), mainly due to the reduction in soil pH and *nosZ* gene abundance, and increase in *nirK* and *nirS* gene abundance and $NO_3^-$-N concentration.

## 4. The Function of Green Manure in the Remediation of Contaminated Paddy Soil

With the development of science and technology and urbanization, pesticides and heavy metals have become widely used, and some soil environments have been polluted through garbage incineration, atmospheric deposition, and sewage and solid waste discharge. Although physical and chemical methods are the main methods of soil pollution control at present, they are difficult and harmful, and should not be widely promoted. Phytoremediation is a strategy based on plants to eliminate or reduce soil and water pollution [85], and includes the use of natural or genetically modified plants to extract harmful substances from the environment, including herbicides, pesticides, heavy metals, radionuclides, etc. These pollutants are converted into safe metabolites, which have the advantages of low cost and lack of damage to the environment and ecology. At present, many studies have been undertaken on environmental phytoremediation, but few reviews have been conducted on the use of functional green manure systems in farmland soil remediation.

*4.1. Remediation of Herbicide Residues*

Herbicides are often used to reduce the consumption of soil nutrients by weeds to maximize the yield of cash crops in agricultural production, but their high residues in the agricultural ecological environment have always been an environmental pollution problem that urgently needs to be solved. Green manure has the characteristics of high rhizosphere activity, fast growth, large biomass, and accumulation of chemical substances in tissues. It can be used for the effective restoration of herbicide residues in paddy soil. Quinclorac

and tebuthiuron are two common herbicides [86]. The former has poor adsorption in soil colloids, resulting in high fluidity in the soil profile. The latter is easily soluble in water, and its leaching depth can reach 50 cm even in soils with high organic matter and clay contents, which seriously threatens the environmental safety of soil, rivers, and groundwater. Green manure remediation involves the absorption and transport of herbicides in the soil solution. Because different green manure species have different sensitivities to herbicides, they may lead to differences in herbicide transport and transformation processes. Mendes et al. [87] used radio labelled herbicide technology to quantitatively analyze the ability of four green manure crops (*Crotalaria spectabilis*, *Canavalia ensiformis*, *Stizolobium aterrimum*, and *Lupinus albus*) to repair quinclorac- and tebuthiuron-contaminated soils, and the results showed that the four green manure crops have significant repairing effects on the two herbicides, and the repairing ability for tebuthiuron (4–22%) is higher than that for quinclorac (2–13%). Similarly, Taliane et al. [88] used the same method to study the effects of six green manures (*Canavalia ensiformes* (L.) *DC.*, *Stilizobium aterrimum* L., *Raphanus sativus* L., *Crotalaria spectabilis RÖth*, *Lupinus albus* L., and *Pennisetum glaucum* (L.) *R. Br.*) on three herbicides (diuron, hexazinone, and sulfometuron-methyl). It was found that all green manures had a repairing effect on the three herbicides, and the green manure *Canavalia ensiformes* (L.) *DC* had the best repairing effect on the herbicide hexazinone. The types of green manures for herbicide restoration are species specific. In addition, green manure use in phytoremediation strategies also significantly shortens the interval between the period of herbicide application and planting of the next crop, thereby reducing the time costs in agricultural production [89].

Previous studies have found that two-year-old legume green manure crops can significantly reduce the weed seed bank of spring sown crops in subsequent years [90], which shows that green manure crops not only help reduce pesticide residues, but also reduce weeds in annual crops after green manure crops are planted, thereby reducing the quantity of herbicides used during rice growth and achieving the goal of protecting the environment.

*4.2. Remediation of Heavy Metals*

The natural concentration of heavy metals in paddy soil depends on the geological characteristics of the soil, but a large number of chemical substances are applied to the soil as fertilizers and pesticides every year, which can lead to a sharp increase in the content of heavy metals in paddy soil. In particular, the accumulation of cadmium (Cd), plumbum (Pb), mercury (Hg), and arsenic (As), and the bioaccumulation capacity and toxicity of these metals in soil, rice, and animal tissues pose a great threat to food and environmental safety [91]. Heavy metals cannot be degraded in the process of phytoremediation and can only be transformed from one organic compound or oxidation state to another organic compound or oxidation state. Phytoremediation of heavy metals changes in their oxidation state to a state with less toxicity, easy volatilization, lower bioavailability, and higher or lower water solubility [92].

Kim et al. [93] discovered that due to rhizosphere acidification and organic chelating agents secreted by rhizosphere and rhizosphere microorganisms, the bioavailability of metals is increased, which has an adverse effect on the growth of rice and other cash crops. Root-induced changes in soil biochemical properties greatly affect the solubility and availability of trace heavy metals [94]. The high specific surface area provided by soil colloids assists in controlling the concentration of heavy metals in natural soil, but this may be metal specific [95]. The application of green manure will increase soil porosity, thereby increasing the specific surface area of the soil and reducing the solubility of heavy metals [96]. In addition, soil aeration, microbial activity, and mineral composition have been shown to affect the availability of heavy metals in the soil. Azimzadeh et al. [97] found that the application of alfalfa green manure can increase the bioavailability of Zn (14%), Cu (26%), and Ni (20%) in the soil, and the absorption of metals increased during the growth of rape plants but decreased during the growth of corn plants in the intercropping of rape and corn. However, Hussain et al. [98] studied the effects of inorganic and organic

fertilizers on the availability and accumulation of heavy metals in the soil and the absorption of rice through long-term field experiments; compared with no fertilizer, the combination of chemical fertilizer with Chinese milk vetch increased the total Cd, Zn, and Cu by 146%, 14%, and 8% in rice grains, respectively. In addition, the combined application of chemical fertilizers and green manures can increase the bioaccumulation factor of Cd.

Green manure crops can also be combined with microbial inoculants to improve the ability of functional restoration plants to absorb heavy metals. Mishra et al. [99] discovered that applying green manure (*Sesbania*), metal dissolving bacteria, and green manure + metal dissolving bacteria respectively increased the solubility of Cd by 34, 123, and 76%; increased the Cd uptake of crops by 125, 175 and 212%; and decreased the Cd activity in the soil by 11, 37 and 42%, with Zn and Pb showing the same trends as that of Cd. Similarly, Omar et al. [100] applied two legume green manures (*Vicia faba var. minor* and *Sulla coronaria*), which had been inoculated with heavy metal resistance substances and plant growth-promoting bacteria into the soil, and found that this type of crop rotation significantly reduced the contents of Cd and Zn in crops, and the concentrations of heavy metals in the soil and improved soil fertility and crop yield.

To explore the mechanism by which green manure repairs Cd pollution, Zhang et al. [101] studied the growth of mustard (*B. napus* tolerant (H18) and sensitive (P9) types) under Cd stress and found that the activity of antioxidant enzymes was the main factor for its resistance, and the high seedling survival rate was mainly controlled by the expression level of the translocator (*BnaHMA4c*) that transports Cd from roots to stems. In addition, exogenous salicylic acid is beneficial to alleviate Cd stress and significantly increases the chlorophyll content and antioxidant enzyme activity of ryegrass, thereby increasing the content of Cd in the cell wall of ryegrass [102]. In short, this kind of green manure crop can not only improve soil fertility but also absorb heavy metals, making this approach an economical and environmentally friendly soil remediation strategy.

## 5. Conclusions and Prospects

Green manure planting has many advantages in rice tillage. This review concludes that winter planting of green manure in paddy fields can reduce soil erosion and improve soil properties, such as soil bulk density, water conductivity, soil porosity, water-holding capacity, enzyme activity, water-stable aggregates, and pH. This review summarizes the climatic adaptations of different green manures. Different fertilizer application methods and different tillage practices have different effects on soil nutrient availability, and mixed fertilizer application methods and no-till rotations are more likely to improve agroecosystems. Green manures vary in their suitability for different soil textures, and milk vetch is suitable for both acidic red and alkaline black soils. Moreover, green manure also has a significant effect on the emissions of the three main greenhouse gases. The application of green manure in combination with fertilizer in the form of mulch to the field reduces $CO_2$ emissions, whereas the C/N ratio of green manure had a significant effect on $CH_4$ and $N_2O$ emissions, with a low C/N ratio reducing $CH_4$ emissions but increasing $N_2O$ emissions, and reducing $N_2O$ emissions from paddy fields during flooding. The planting of green manure reduces herbicide residues and suppresses weeds in paddy fields and enriches and transfers specific heavy metals, such as Cd, Pb, and Zn, thereby remediating the soil. During the growth of different green manure crops, the improvement in soil properties and nutrient supply vary with the growth stage. Regrettably, this review does not provide a comprehensive quantitative investigation of the contribution of green manure to nutrient availability, greenhouse gas emissions, and soil remediation in paddy systems. Therefore, the regional adaptability of green manure tillage methods should be different. It is necessary to optimize the choice of green manure and return methods, and to quantify the role of green manure according to the specific green manure species and soil texture, so that the green manure tillage pattern can be rationalized.

Although the varieties of green manure are diverse, there is no assessment of their effect on specific soil textures or rice habitats. Thus, it is important to strengthen the

collection of existing green manure resources and to quickly obtain new varieties with stable traits through molecular techniques and cross breeding. The role of green manure in increasing nutrients, reducing gas emissions, and soil remediation remains highly variable, and there is a necessity to study the mechanisms of nutrient transfer from green manure to rice, to thus quantify the nutrient supply of green manure to paddy systems, and to quantify the contribution of green manure to greenhouse gas emissions and soil remediation in paddy fields. In addition, drawing on the concept of creative agriculture, the use of green manure in winter can be applied to the service, tourism, and other leisure industries, thereby increasing the economic benefits of green manure cultivation.

**Author Contributions:** H.Y. and J.W. contributed to the conception and design. B.L. wrote the manuscript. H.Y. and J.W. contributed to the revision of the manuscript and secured funding. All authors have read and agreed to the published version of the manuscript.

**Funding:** This research was funded by the Young Scientists Fund of the National Natural Science Foundation of China (Grant No.41701282).

**Conflicts of Interest:** The authors declare no conflict of interest.

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
