# Peer review of "Ecological and Environmental Benefits of Planting Green Manure in Paddy Fields"

_agriculture, doi:10.3390/agriculture12020223_

Round 1
Reviewer 1 Report
The present paper „Ecological and Environmental Benefits of Planting Green Manure in Paddy Field: A Review“ is very interesting paper that definitely fits into scope of Agronomy journal. Draft provides the new insight of agricultural practice of green manure on soil quality and climate regulation. The review shows proper use of numerous sources, but some part I do not see enough precise. When improving please pay attention on details. Too many general statements occur in the draft. Please provide to readers the information of type of green manure and textural type of soil where this green manure is incorporated in soil.
Other weak parts are mainly related to little bit clumsy choice of terms (e.g. see the suggestions in Abstract), or much shallowed presented the part of the sections. E.g., the physical and chemical properties part is very vague. Here I suggest to authors that present current state of the art of green manure on soil physical and chemical properties. Provide new insight how green manure crops mitigate erosion? Connect this with infiltration and filtration. In addition, this also with differential porosity and soil structure. All this should be connected with textured soil type and type (and duration) of green manure practice in particular environment. Finally, to this subsection, soil chemical properties cannot refer only to soil pH, especially when authors only mention the “organic matter, total nitrogen and available nitrogen contents” as enumeration.
Nutrient utilisation subsection should have more deeply knowledge about other soil nutrients than N.
Part of conclusion section did not studied in main text. Please carefully read again and make a revision.
Minor corrections
Line 14. “Adequate” - Superfluous. Please delete.
Line 17. I believe that “fertilizer” here is not the best word. Try with “nutrients” instead.
Line 19. This is disconnected sentence with previous and following one.
Line 20. Here and elsewhere, please avoid using "we".
Line 31. “fertilizer sources” - substitute with nutrients please.
Line 39. “fertility” - nutrients. Fertility is much wider term and do not refer only to chemical substances.
Line 41. What this means? Do you mean "rapid occurrence of soil compaction". This is very un-precise. How the mineral fertilizers can speed the compaction problems. Why you do not rather mention that that application of only mineral fertilizer distort soil structure and soil become vulnerable to wheeling.
Line 41-42. Please avoid general writing. Please be specific. Define "other environmental problems"
Line 45. I believe you want to tell here "agricultural land" instead "agricultural field"?
Line 64-76. In review papers, it is advisable to provide for readers the type of green manure and soil texture type where this manure was incorporated.
Line 95. “fertilizer” - Please be consistent in paper. Substitute with "green manure"
Line 98-101. This sentence is not clear. Please clarify which treatments you compare here. Green manure with fallow, but in some crop rotation? Please be specific.
Line 106. “Green fertilizer” - please unify terms.
Line 132-133. Please be specific. Which green manure crops have large C/N ration. The leguminous ones or non-leguminous ones?.
Line 136. Please check again. I believe that Ca and Mg are not trace elements
Line 172. Please use mineral fertilizer as term instead.
Line 180. Please substitute with residue. There are green manure species that do not have straw as residue.
Line 190-191. Please add supportive reference here.
Line 216-218. This should have reference.
Line 290. “non cash crops” - please substitute this with "weeds"
Line 342-344. Please add supportive reference
Line 377. “Conservation tillage” - this subject almost that is not mentioned in this review. Please avoid conclusions derived from non-presented subject.
Reviewer 2 Report
The authors reviewed the effects of green manure on the soil microenvironment and greenhouse gas emissions and the role of green manure in phytoremediation of paddy fields. They concluded that green manure could not only affect soil nutrients and the microbial community but also reduce greenhouse gas emissions and enhance soil remediation to some extent. This paper indicates useful information for the green and sustainable development of rice planting. However, some modifications for more explanation will be required.
Specific comments:
1) Abstract: Line 24: The authors mentioned that the related germplasm resources and farming patterns are discussed. However, no detail discussion of the germplasm resources and farming patterns in this paper. Please clarify it.
2) Abstract: Line 25: What is theoretical guidance for the green and sustainable development of rice planting? Please make clear.
3) P.4, line 1: What is dry matter accumulation? It is better to explain it.
4) p.5, line 212-213: Please explain what is conventional no-tillage paddy fields and powder-ridge no-tillage paddy fields.
5) Conclusions: The authors discussed on the effects on greenhouse gas emission in Chapter 3 and the remediation of contaminated paddy soil in Chapter 4. However, these are few explanation in conclusion. Please summarize main discussion points in this paper.
Round 2
Reviewer 1 Report
Authors made a major revision and significantly improve their work. New version of the paper not contain any weakness which appear in first draft. Answers are correct and extensive. I do not have any raised questions except few typo errors which occurs mainly in new added text in second draft.
Kind regards
